# Low Replication Efficiency of a Japanese Rabbit Hepatitis E Virus Strain in the Human Hepatocarcinoma Cell Line PLC/PRF/5

**DOI:** 10.3390/v15061322

**Published:** 2023-06-05

**Authors:** Wenjing Zhang, Milagros Virhuez Mendoza, Yasushi Ami, Yuriko Suzaki, Yen Hai Doan, Ken Maeda, Tiancheng Li

**Affiliations:** 1Department of Virology II, National Institute of Infectious Diseases, Gakuen 4-7-1, Musashi-murayama, Tokyo 208-0011, Japan; zwjviolin@foxmail.com; 2Department of Veterinary Science, National Institute of Infectious Diseases, Toyama 1-23-1, Shinjuku-ku, Tokyo 162-8640, Japan; mvirhuez@niid.go.jp (M.V.M.); kmaeda@niid.go.jp (K.M.); 3Division of Experimental Animals Research, National Institute of Infectious Diseases, Tokyo 208-0011, Japan; yami@nih.go.jp (Y.A.); ysuzaki@niid.go.jp (Y.S.); 4Center for Emergency Preparedness and Response, National Institute of Infectious Diseases, Tokyo 208-0011, Japan; yendoan@niid.go.jp

**Keywords:** hepatitis E virus, rabbit HEV, JP-59 strain, PLC/PRF/5 cell, persistent infection

## Abstract

A Japanese rabbit hepatitis E virus (HEV) strain, JP-59, has been identified in a feral rabbit. When this virus was transmitted to a Japanese white rabbit, it caused persistent HEV infection. The JP-59 strain shares an <87.5% nucleotide sequence identity with other rabbit HEV strains. Herein, to isolate JP-59 by cell culture, we used a 10% stool suspension recovered from a JP-59-infected Japanese white rabbit and contained 1.1 × 10^7^ copies/mL of the viral RNA and using it to infect a human hepatocarcinoma cell line, PLC/PRF/5. No sign of virus replication was observed. Although long-term virus replication was observed in PLC/PRF/5 cells inoculated with the concentrated and purified JP-59 containing a high titer of viral RNA (5.1 × 10^8^ copies/mL), the viral RNA of JP-59c that was recovered from the cell culture supernatants was <7.1 × 10^4^ copies/mL during the experiment. The JP-59c strain did not infect PLC/PRF/5 cells, but its intravenous inoculation caused persistent infection in rabbits. The nucleotide sequence analyses of the virus genomes demonstrated that a total of 18 nucleotide changes accompanying three amino acid mutations occurred in the strain JP-59c compared to the original strain JP-59. These results indicate that a high viral RNA titer was required for JP-59 to infect PLC/PRF/5 cells, but its replication capability was extremely low. In addition, the ability of rabbit HEVs to multiply in PLC/PRF/5 cells varied depending on the rabbit HEV strains. The investigations of cell lines that are broadly susceptible to rabbit HEV and that allow the efficient propagation of the virus are thus needed.

## 1. Introduction

The Hepatitis E virus (HEV) is a positive-sense single-strand RNA virus that is classified in the family *Hepeviridae*, which includes the two subfamilies: *Orthohepevirinae* and *Parahepevirinae* (ictv.global/taxonomy). The subfamily *Orthohepevirinae* includes at least four genera: *Paslahepevirus*, *Rocahepevirus*, *Chirohepevirus*, and *Avihepevirus* [1]. The genus *Paslahepevirus* was classified into two species, *alci* and *balayani*, and the species *balayani* included eight viral genotypes, HEV-1 to HEV-8 [1]. It has been confirmed that HEV-1 to HEV-4, HEV-7, and rat HEV transmit to humans and cause hepatitis E and rat HEV belonging to the genus *Rocahepevirus* [2,3,4].

Rabbit HEV was originally identified in farmed rex rabbits and is genetically close to HEV-3 [5,6]. However, rabbit HEV shares an <80% nucleotide sequence identity with other HEV-3 strains and forms a distinct group; rabbit HEV is provisionally assigned as a subtype of HEV-3, i.e., HEV-3ra, by the International Committee on the Taxonomy of Viruses (ICTV) Hepeviridae Study Group [7,8]. Rabbit HEV contains approx. 7.2-kb RNA genomes encoding three discontinuous and partially overlapped open reading frames (ORFs). ORF1 encodes a large non-structure polyprotein with several putative functional proteins. ORF2 encodes the major capsid protein. ORF3 encodes a phosphoprotein with multiple functions [5,9].

To date, a large number of rabbit HEV strains have been detected worldwide in farmed rabbits, wild rabbits, and specific pathogen-free rabbits, suggesting that (i) virus infection is common in rabbits and (ii) rabbits are a natural host of HEV [10,11,12,13,14,15,16]. Cell culture for growing rabbit HEV has been established using the human hepatocarcinoma cell line PLC/PRF/5 and the human lung carcinoma cell line A549 [17], and infectious viruses have been produced by a reverse genetics system [18].

Rabbit HEV is also capable of infecting pigs and cynomolgus monkeys under experimental conditions, suggesting that this virus has a potential for cross-species transmission [19,20]. In fact, rabbit HEV is known to transmit to humans and causes zoonotic infection, and numerous cases of zoonotic infections in humans have been reported in France, Switzerland, and Ireland [21,22,23].

A rabbit HEV strain, JP-59, was identified in a feral rabbit in Japan [16]. It was confirmed that the strain JP-59 shared an <87.5% nucleotide sequence identity with other known rabbit HEVs and caused persistent infection when it was transmitted to a Japanese white rabbit [16]. In the present study, we characterized this strain JP-59 by inoculating PLC/PRF/5 cells with the virus. We observed that although JP-59 replicated in the cells, the virus titer was lower than that of rbIM223LR: a rabbit HEV strain that was originally identified in a farmed rabbit in Inner Mongolia, China, and then produced by a reverse genetic system, suggesting that the replication ability of rabbit HEV in cell cultures varies depending on the strain.

## 2. Materials and Methods

### 2.1. Rabbit HEV Strains and Preparation of the Virus Suspensions

A rabbit HEV strain JP-59 (GenBank accession no. LC535077) was identified in a feral rabbit in 2018 in Japan and isolated by transmitting it to a Japanese white rabbit [16]. Another rabbit HEV strain, rbIM223LR (accession no. LC484431), was identified in 2011 in Inner Mongolia, China, and infectious viruses were produced by a reverse genetics system [17,18]. Both JP-59 and rbIM223LR caused persistent infection when administered to Japanese white rabbits. We, thus, used rbIM223LR as a positive control in this study.

A total of 5 g of the fecal specimens collected from a JP-59-infected rabbit [16] on day 520 post-infection (p.i.) were diluted with 50 mL of a 10 mM phosphate-buffered saline (PBS) to prepare a 10% (*w*/*v*) suspension, which was shaken at 4 °C for 1 h, clarified by centrifugation at 10,000× *g* for 30 min, and then passed through a 0.45-µm membrane filter (Millipore, Bedford, MA, USA). This suspension contained 1.1 × 10^7^ copies/mL of viral RNAs, and we named it JP-59s.

The fecal specimens derived from a rbIM223LR-infected rabbit on day 680 p.i. were treated similarly, adjusted to the same RNA copy number, and we named the suspension “LRs”. The final suspensions were stored at −80 °C until their use for the inoculation of PLC/PRF/5 cells.

### 2.2. Purification of JP-59 from Stool Suspension

To raise the viral titer and remove potential virus inhibitors and toxic substances from the fecal specimens, a total of 200 g of the fecal specimens was collected from a JP-59-infected rabbit [16] on days 736–738 p.i. and a 10% stool suspension was prepared as described above. This suspension was concentrated by centrifugation at 126,000× *g* for 3 h at 4 °C in a Beckman SW32Ti rotor (Beckman Coulter. Inc. Brea, CA, USA), and the pellet was suspended with PBS. The virus was purified by CsCl gradient centrifugation. Briefly, 4.5 mL of the samples were mixed with 2.1 g of CsCl and centrifuged at 100,000× *g* for 24 h at 10 °C in the Beckman SW55Ti rotor. This gradient was fractionated into 250-µL aliquots, and the viral RNA in each fraction was examined by a quantitative real-time reverse transcription-polymerase chain reaction (RT-qPCR).

As shown in Figure 1, a peak of the viral RNA appeared in fraction 5; the copy number was 5.2 × 10^9^ copies/mL. The copy number in fraction 6 was 1.3 × 10^9^ copies/mL. For the removal of CsCl, fractions 5 and 6 were diluted with PBS and centrifuged at 100,000× *g* for 2 h at 4 °C in the Beckman SW55Ti rotor (Beckman). The pellets were resuspended in 1.5 mL of PBS and passed through a 0.45-µm membrane filter (Millipore). We named the viruses purified from fractions 5 and 6 JP-59F5 and JP-59F6, respectively. As shown by RT-qPCR, the final RNA copy number of JP-59F5 was 5.1 × 10^8^ copies/mL, and that of JP-59F6 was 1.2 × 10^8^ copies/mL.

### 2.3. Cell Culture and Virus Infection

The human hepatocarcinoma cell line PLC/PRF/5 (JCRB0406) and the human lung carcinoma cell line A549 (IF0505153) were obtained from the Health Science Research Resources Bank and grown in Dulbecco’s modified Eagle’s medium (DMEM) supplemented with 10% (*v*/*v*) heat-inactivated fetal bovine serum (FBS; Nichirei Biosciences, Tokyo, Japan), 100 U penicillin, and 100 mg streptomycin (Gibco, Grand Island, NY, USA) at 37 °C in a humidified 5% CO_2_ atmosphere. For virus inoculation, the confluent cells were trypsinized and cultured in a 25-cm^2^ culture bottle (5 × 10^5^ cells/bottle).

The next day, PLC/PRF/5 cells were inoculated with stool suspension or purified viruses and were adsorbed at 37 °C for 1 h. The cells were washed three times with PBS before being supplemented with a 10 mL maintenance medium consisting of medium 199 (Invitrogen, Carlsbad, CA, USA) containing 2% (*v*/*v*) heat-inactivated FBS and 10 mM MgCl_2_. The incubation was conducted at 36 °C, and the culture medium was replaced with a new medium every 4 days.

### 2.4. Inoculation of Rabbits and Sample Collection

The cell culture supernatant containing 7.1 × 10^4^ copies/mL of the viral RNA was collected from JP-59F5-infected PLC/PRF/5 cells on day 60 p.i., clarified by centrifugation at 10,000× *g* for 30 min, and then passed through a 0.45-µm membrane filter (Millipore). Two 1-year-old female Japanese white rabbits (Rab59-1 and Rab59-2) (SLC, Shizuoka, Japan) were intravenously inoculated with 1 mL of the supernatant through an ear vein.

Serum samples were collected weekly from the rabbits and used for the detection of rabbit HEV RNA, anti-rabbit HEV IgG antibodies, and alanine aminotransferase (ALT) values. Fecal specimens were collected every 3 or 4 days, and 10% (*w*/*v*) stool suspensions were used for the detection of the viral RNA.

The animal experiments were reviewed and approved by the institutional ethics committee and were performed according to the ‘Guides for Animal Experiments at the National Institute of Infectious Diseases, Tokyo, Japan, under codes 119083 (15 August 2019) and 119149 (9 January 2020). The rabbits were individually housed in Biosafety Level-2 facilities. They were negative for anti-HEV IgG antibodies and rabbit HEV RNA before inoculation.

### 2.5. Detection of Rabbit HEV RNA

Viral RNA was extracted from 200 µL of the cell culture supernatants or 10% stool suspensions by a MagNA Pure 96 System (Roche Applied Science, Mannheim, Germany) with a MagNA Pure 96 DNA and Viral NA Small Volume Kit (Roche Applied Science), and the copy numbers were examined by a one-step RT-qPCR using a TaqMan Fast Virus 1-step Master Mix (Applied Biosystems, Foster City, CA, USA) and a QuantStudio 3 Real-Time PCR System (Applied Biosystems) as described [24]. A 10-fold serial dilution (10^7^ to 10^1^ copies) of the synthetic entire genome RNA of rbIM223LR was used as the standard for the quantitation of the viral genome copy numbers.

The partial genome of the viral RNA was amplified by a nested RT-PCR with nine sets of primers (Table 1). Both the first and second PCR amplifications were carried out under the following conditions: inoculation at 96 °C for 60 s, followed by 35 cycles of 30 s at 95 °C, 30 s at 52 °C, and 60 s at 72 °C, and a final extension at 72 °C for 7 min. The PCR products were purified using a QIAquick PCR purification kit (Qiagen, Hilden, Germany), and nucleotide sequencing was carried out with the primers used for the second PCR. 

### 2.6. Detection of Rabbit Anti-HEV IgG Antibodies

The IgG antibodies against rabbit HEV were detected by an in-house enzyme-linked immunosorbent assay (ELISA) using virus-like particles (VLPs) of rabbit HEV as the antigen. The cutoff value for the anti-HEV IgG was 0.151 [25].

### 2.7. Liver Enzyme Level

ALT values in the sera from rabbits were monitored weekly using a Fuji Dri-Chem Slide GPT/ALT-PIII kit (Fujifilm, Saitama, Japan) according to the manufacturer’s recommendations.

### 2.8. Next-Generation Sequencing (NGS) for the Entire JP-59 Genome

The fecal suspension and the infected cell culture supernatant were concentrated by ultracentrifugation at 126,000× *g* for 3 h in a Beckman SW32Ti rotor (Beckman), and the pellet was resuspended with PBS and used for NGS. The entire genome sequence of JP-59 was determined by NGS as described [26]. Briefly, viral RNA was extracted from 200 µL of the 10% stool suspensions, and a 200-base pair (bp) fragment library was constructed for each sample with an NEBNext Ultra RNA Library Prep Kit for Illumina ver. 1.2 (New England Biolabs, Ipswich, MA, USA) according to the manufacturer’s instructions. Samples were bar-coded for multiplexing with the use of NEBNext Multiplex Oligos for Illumina and Index Primer Sets 1 and 2 (New England Biolabs). Library purification was conducted with Agencourt AMPure XP magnetic beads (Beckman Coulter, Pasadena, CA, USA) as recommended in the NEBNext protocol. A 151-cycle paired-end read sequencing run was carried out on a MiSeq desktop sequencer (Illumina, San Diego, CA, USA) using the MiSeq Reagent Kit ver. 2 (300 cycles).

## 3. Results

### 3.1. No Virus Was Recovered from PLC/PRF/5 Cells Inoculated with JP-59s

To isolate the JP-59 by using PLC/PRF/5 cells, we inoculated the cells with 1 mL of JP-59s: a stool suspension derived from a persistently JP-59-infected rabbit that contained 1.1 × 10^7^ copies/mL of viral RNA. Triplicate samples were used for the inoculation. In total, with 1 mL of LRs, the virus suspension derived from a rbIM223LR-infected rabbit contained the same RNA copy number and was used as a positive control. The viral RNA in the LRs-infected cells was first detected on day 4 p.i. at 1.21 × 10^4^ copies/mL; it decreased to 3.12 × 10^3^ copies/mL on day 8 p.i. and then increased gradually, reaching 1.3 × 10^7^ copies/mL on day 140 p.i. By contrast, the viral RNA in the supernatants of JP-59s-infected cells decreased rapidly and became undetectable after 20 days p.i., although viral RNA was detected on day 4 p.i. at 2.29 × 10^4^ copies/mL (Figure 2). This observation was performed until day 148 p.i., and no evidence of virus replication was observed.

A549 cells have been confirmed to be susceptible to rabbit HEV including Inner Mongolia isolates such as rbIM214L and rbIM223L [17]; however, we observed no evidence of virus replication in the case of JP-59. These results indicate that direct inoculation of a 10% stool suspension of the virus onto the existing susceptible cells is not sufficient for the further propagation of the JP-59s strain.

### 3.2. JP-59 Replicates in PLC/PRF/5 Cells When a Purified High-Titer Virus Was Used

In an attempt to investigate whether the purified high-titer JP-59 infected PLC/PRF/5 cells, we inoculated JP-59F5 or JP-59F6 onto the cells and monitored the viral RNAs (Figure 3). In JP-59F5-infected cells, viral RNA was detected in the culture supernatant on day 4 p.i., and the copy number was 3.7 × 10^4^ copies/mL. The RNA gradually decreased to 5.4 × 10^3^ copies/mL on day 28 p.i. and increased gradually from day 32 p.i., reaching 5.0 × 10^4^ copies/mL on day 40 p.i. and staying at a plateau between 2.5 × 10^4^ copies/mL and 7.1 × 10^4^ copies/mL until day 120 p.i. We named the virus recovered from the supernatants of JP-59F5-infected cells JP-59c. In the case of JP-59F6, the viral RNAs showed a similar pattern until day 60 p.i., although these copy numbers were lower than those of JP-59F5.

For the determination of whether the cell passage affected the viral replication, JP-59F6-inoculated cells were trypsinized and passaged at day 60 p.i. The cells were washed three times with PBS and incubated with 1 mL of 0.05% trypsin at 37 °C for 10 min. After the trypsin was removed, the cells were diluted with a 20 mL maintenance medium, separated into two bottles and incubated at 36 °C. The viral RNA in the supernatants rapidly decreased after the passage and became undetectable on day 8 post-passage.

These results indicated that the JP-59 strain grew in PLC/PRF/5 cells if the purified high-titer virus was used and a constant growth condition was maintained. However, the maximum copy number of the viral RNA was >100 times lower than that of the control rbIM223LR strain.

### 3.3. Infectivity of JP59c

To examine the infectivity of JP-59c recovered from the cell culture, we prepared two samples for inoculation: (i) an un-concentrated supernatant, i.e., the supernatant of JP-59F5-inoculated cells collected on day 60 p.i., containing 7.1 × 10^4^ copies/mL of the viral RNA, and (ii) a concentrated supernatant, in which a total of 110 mL of the supernatants collected from day 40 to 120 p.i. were mixed and concentrated at 126,000× *g* for 3 h in a Beckman SW32Ti rotor (Beckman), and this pellet was then suspended in 1.5 mL of PBS and then passed through a 0.45-µm membrane filter (Millipore). The concentrated supernatant contained 1.8 × 10^6^ copies/mL of viral RNA. However, both the un-concentrated and concentrated supernatants did not show any sign of replication upon their infection to PLC/PRF/5 cells.

For the examination of the in vivo infectivity of JP-59c, two Japanese white rabbits (Rab59-1 and Rab59-2) were inoculated with 1 mL of the supernatant containing viral RNA at 7.1 × 10^4^ copies/mL. After inoculation via the ear vein, the viral RNA was detected in the feces of Rab59-1 on day 7 p.i. at 3.76 × 10^3^ copies/g; it gradually increased to 7.85 × 10^7^ copies/g on day 42 p.i. and was then maintained at approx. 1.01 × 10^6^ to 2.96 × 10^7^ copies/g until day 133 p.i. We named the virus recovered from Rab59-1’s fecal specimens JP-59cr.

In the case of Rab59-2, the viral RNA was first detected on day 21 p.i. at 1.59 × 10^4^ copies/g; it had gradually increased to 7.82 × 10^6^ copies/g on day 53 p.i. and then gradually decreased but was detected at approx. 1.12 × 10^4^ copies/g until day 133 p.i. (Figure 4a). The viral RNA in all serum samples was under detection limits by the RT-qPCR.

The IgG antibody was first detected on day 28 in Rab59-1 with an optical density (OD) value of 0.429 and on day 56 p.i. in Rab59-2 with an OD value of 0.161. The IgG peaked on day 49 p.i. in Rab59-1 with an OD value of 3.309 and then stayed at similar levels until day 133 p.i. In contrast, the peak value of the IgG was 0.601 in Rab59-2 on day 91 p.i. (Figure 4b). Although the ALT levels in these two rabbits differed, no significant elevation was observed during the infection period (Figure 4c). These results indicated that the JP-59c isolated by the cell culture was infectious in vivo.

### 3.4. Nucleotide Sequence Analyses of JP-59 Strains

The entire genome sequences of JP-59F5, JP-59F6, and JP-59cr were analyzed by NGS, which revealed that all the viruses consisted of 7282 nucleotides plus a poly (A) tail with an undetermined length. The 5’ terminal untranslated region (5’UTR) and 3’UTR contained 26 nucleotides (nucleotide sequence nos. 1–26) and 70 nucleotides (7213–7282), respectively. ORF1 (27–5195) encoded 1723 amino acids (aa), ORF3 (5192–5560) encoded 122 aa, and ORF2 (5230–7212 nt) encoded 660 aa.

JP-59F5 and JP-59F6 were collected on days 736–738 (approx. 2 years) p.i., and both entire genome sequences were identical to that of the original JP-59 collected from the same rabbit at 12 weeks p.i. [16], suggesting that JP-59, which replicated in the rabbit, was genetically stable. In contrast, a total of 18 nucleotide changes (15 in ORF1, two in ORF2, and one in ORF3) were identified in JP-59cr (Table 2). These nucleotide changes were accompanied by two aa mutations (Thr566Ile and Gly749Ser) in ORF1 and one aa mutation (Ser4Phe) in ORF3 (Table 2). The mutation Thr566Ile was observed in the hepevirus unique domain and Gly749Ser is located in the hypervariable polyproline region of ORF1. The amino acid, 566Ile, was identified in rbIM223LR, but 749Ser and 4Phe were not present in rbIM223LR. No mutation was found in the 5’UTR or 3’UTR.

Since the virus titer of JP-59c was quite low, the entire genome sequence could not be determined by NGS. For the confirmation of whether the mutation occurred during the replication in cell culture, nine partial genome fragments of JP-59c that covered all the positions of these 18 mutations were amplified by RT-PCR using the primers shown in Table 1. The nucleotide sequence analyses confirmed that all 18 mutations were present in JP-59c, demonstrating that these mutations in JP-59cr were derived from JP-59c and occurred during replication in PLC/PRF/5 cells. 

## 4. Discussion

Although a cell culture system that multiplied rabbit HEV was established using virus strains such as rbIM214L and rbIM223L, which were isolated in Inner Mongolia [17], it is unknown whether other strains were capable of replicating in the cell culture. JP-59 is the only rabbit HEV strain identified in Japan [16]. To establish a cell culture for growing this strain, we inoculated the virus onto PLC/PRF/5 cells because the PLC/PRF/5 was broadly susceptible to HEV infection, and the cell culture of rabbit HEV was established using this cell line. However, the ability of JP-59 to replicate in the cell culture was significantly lower than that of an Inner Mongolia isolate, rbIM223LR, suggesting that the ability of rabbit HEV to propagate in the cell culture varied. In other words, PLC/PRF/5 cells were not the best choice for this purpose, and a novel cell culture system for rabbit HEV was required. It is of interest to examine whether the cell lines derived from rabbits were susceptible to rabbit HEV infection.

We observed the replication of JP-59 in the cells only when the virus solution containing >10^8^ copies/mL of the viral RNA was used, demonstrating that a high titer of JP-59 was required to initiate an infection in PLC/PRF/5 cells. Although the cell culture is a convenient method for HEV isolation, it was difficult to maintain the original viral sequence by the cell culture due to the unexpected mutations that often occurred during the long-term virus multiplication in the cell culture. Our findings demonstrate that it is difficult to isolate viruses with a low replication capacity, such as strain JP-59. In contrast, we observed that JP-59 recovered from the cell culture supernatants, which had low copy numbers of the viral RNA, could infect rabbits, suggesting that the isolation of the virus by injecting rabbits is another option for rabbit HEV isolation. Since the JP-59 that was replicated in the rabbits was genetically stable, these rabbits might provide a useful animal model to obtain rabbit HEV with the original sequence.

The nucleotide acid sequence identity between the JP-59 and rbIM223LR strains was 87.4%, and it is of interest to identify the nucleotide acid and amino acid changes that are responsible for better rabbit HEV replication.

## Figures and Tables

**Figure 1 viruses-15-01322-f001:**
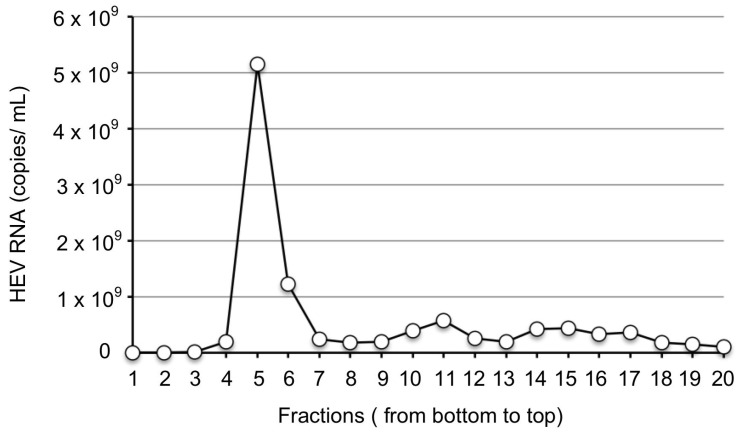
Purification of the JP-59 viruses. Fecal specimens were collected from the JP-59-infected rabbit on days 736–738 p.i., and a 10% suspension was prepared. The virus was concentrated by an ultracentrifugation and then purified by a CsCl equilibrium density gradient ultracentrifugation. The copy numbers of the virus RNA in each fraction were determined by RT-qPCR (○).

**Figure 2 viruses-15-01322-f002:**
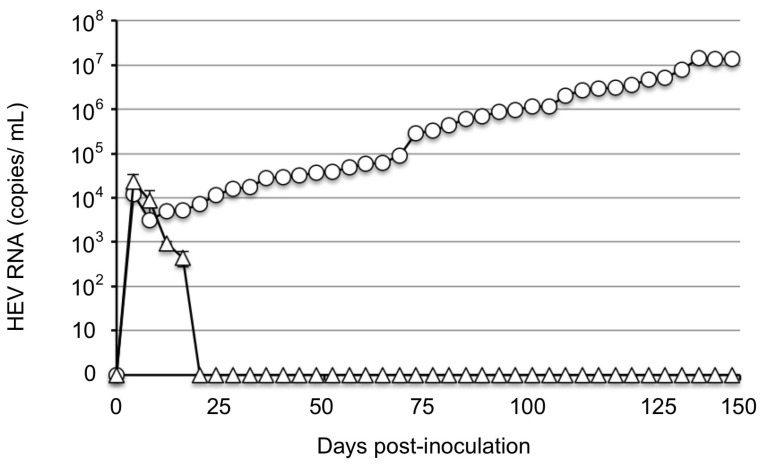
The replication of strains JP-59 and rbIM223LR in PLC/PRF/5 cells. PLC/PRF/5 cells were cultured in 25-cm^2^ bottles. The stool suspensions (JP-59s) derived from a JP-59-infected rabbit containing 1.1 × 10^7^ copies/mL of the viral RNA were inoculated onto three bottles of PLC/PRF/5 cells (△). The stool suspension (LRs) derived from the rbIM223LR-infected rabbit containing the same copy number of the viral RNA was inoculated onto one bottle of PLC/PRF/5 cells and used as a positive control (○). The culture supernatants were collected every 4 days and used for the detection of viral RNA by RT-qPCR.

**Figure 3 viruses-15-01322-f003:**
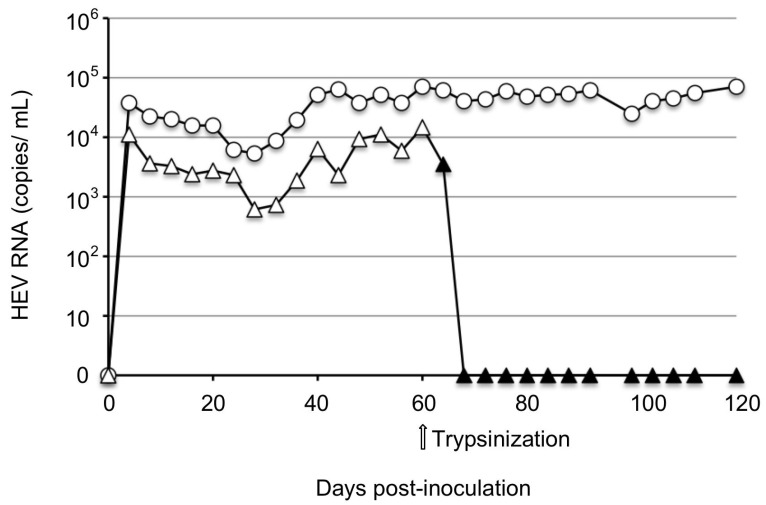
The replication of strain JP-59 in PLC/PRF/5 cells. One milliliter of the concentrated purified JP-59F5 (○) and JP-59F6 (△), respectively, was inoculated onto PLC/PRF/5 cells. The JP-59F6-infected cells were harvested, trypsinized, and passaged on day 60 p.i. and maintained similarly thereafter (▲). Arrow: the time point of the trypsinization. The culture supernatants were collected every 4 days and used for the detection of viral RNA by RT-qPCR.

**Figure 4 viruses-15-01322-f004:**
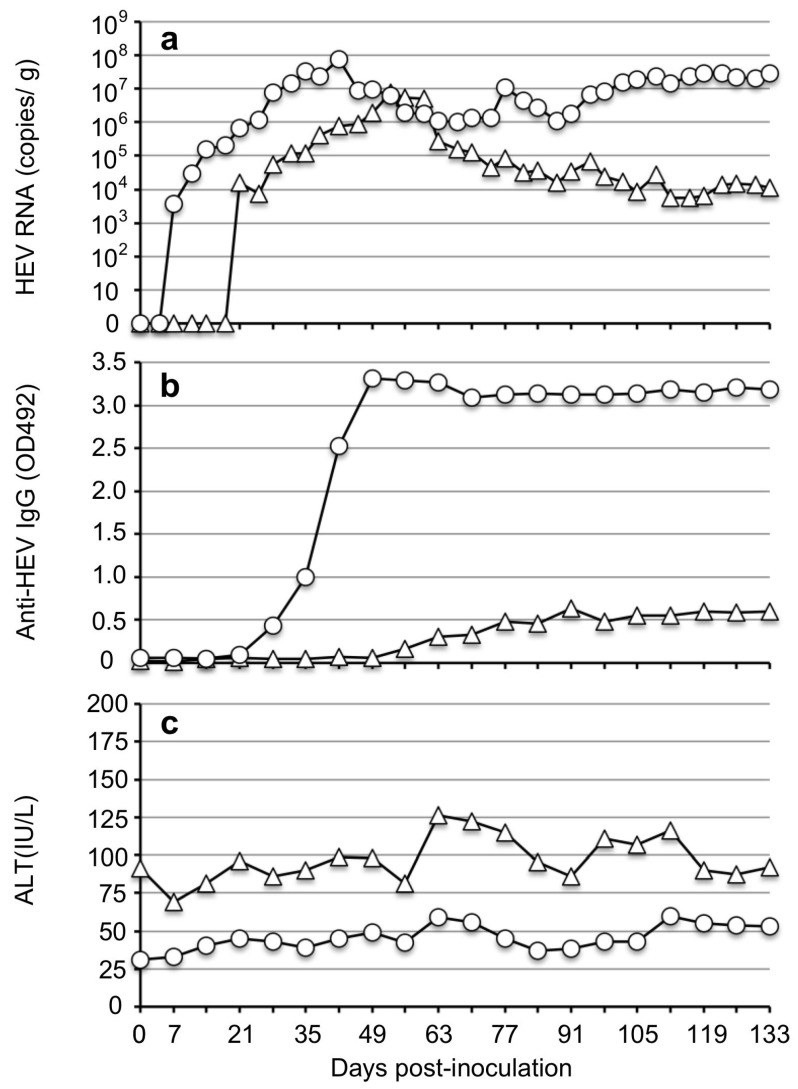
The infectivity of strain JP-59c in vivo. Two Japanese white rabbits, Rab59-1 (○) and Rab59-2 (△) were intravenously inoculated with the cell culture supernatants containing 7.1 × 10^4^ copies/mL of the viral RNAs (JP-59c), which were derived from the JP-59F5-inoculated cells on day 60 p.i. The fecal specimens were collected every 3 or 4 days, and the serum samples were collected weekly. The kinetics of the viral RNA in the feces were determined by RT-qPCR (**a**). The serum anti-rabbit HEV IgG antibodies were determined by ELISA (**b**). ALT values were measured by a commercial kit (**c**).

**Table 1 viruses-15-01322-t001:** Primers used to amplify the partial genome of JP-59c.

Name	For the First PCR	Name	For the Second PCR	Products, nt
F38	TCGATGCCATGGAGGCCCA	F105	GCCTTGGCGAATGCTGTGG	105–469
R560	CATTGCCTCTGCCACATCAG	R469	TCAAAGCAATAGGTTCGATC	
F373	GCGATGGTATTCTGCCCCTA	F540	CTGATGTGGCAGAGGCAATG	373–1100
R1128	CATCTTCAGAGGCATTCCAG	R1110	AGCTACAAGAGCACCAACAG	
F1337	GCTTTCTGCCGGCTTTCATT	F1378	TTGACGAATCTGTGCACTGC	1378–1759
R1805	CTGAGCAGTACGTTCTCTCA	R1759	TCAGTGATTGTGGTGCGGAA	
F2089	GGGAGTCTACTAACCCGTTT	F2160	AGCGACTTTTCACCACCTGA	2160–2439
R2519	CTTGGTATGGTCGAAAGACT	R2439	ATGCGTTAACCAGCCAGTCA	
F2501	GTCTTTCGACCATACCAAGT	F2581	TCGCCCCTGATTATAGAGTT	2581–3040
R3074	CTGCTGTATGGACCTCGATT	R3040	AACGCCTGCAGTGAATTGATA	
F3221	CCACCTGCTTTTGTTGCATA	F3301	TTGAGCATGCCGGTCTAGTC	3301–3699
R3840	GATAGGCACTAATCTGGCAG	R3699	ATATGCCCACCTCACGAAGT	
F4048	GTATGGGCGTCGAACAAAGT	F4141	CTACAACGTGCGAGCTTTAT	4141–4359
R4440	CAGAGTCTTCATAGGCGTCA	R4359	GCCCAAATAAGGCGCAGAAT	
F5020	CGAATGTTGCTCAGGTGTGT	F5091	CTTATTGGCATGTTACAGAC	5091–5440
R5280	CAGCATAGGCAAACACACGA	R5440	CGGAATGTGAGTCAACGTCA	
F5986	TATCGTAACCAGGGGTGGCG	F6022	GGTGTAGCCGAGGAGGAGGC	6022–6650
R6702	AGGGTTAGTTGACGAGCCAT	R6650	GAGAGCCACAACACATCGTT	

**Table 2 viruses-15-01322-t002:** Nucleotide and amino acid changes in rabbit HEV recovered from a JP-59c-infected rabbit.

Nucleotide Changes			Amino Acid Changes			
^a^ Position	JP-59	JP-59cr	^b^ Position	ORF1	ORF2	ORF3
249	C	T				
368	C	T				
551	G	A				
800	T	C				
1040	G	A				
1439	C	T				
1613	A	G				
1652	G	A				
1723	C	T	566	Thr/Ile		
2261	C	T	749	Gly/Ser		
2271	G	A				
2690	C	T				
2930	C	T				
3455	T	C				
4304	C	T				
5205	C	T	4			Ser/Phe
6351	C	T				
6594	A	G				

^a^ Nucleotide positions in the genomic RNA. ^b^ Amino acid positions in each ORF.

## Data Availability

The sequences of HEV used in the study were assigned (GenBank accession nos. LC535077 and LC484431).

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
