# Peer review of "Low Replication Efficiency of a Japanese Rabbit Hepatitis E Virus Strain in the Human Hepatocarcinoma Cell Line PLC/PRF/5"

_viruses, 2023, doi:10.3390/v15061322_

Round 1
Reviewer 1 Report
Comments and Suggestions for Authors
Zhang et al. reported that the low replication efficacy of a Japanese Rabbit HEV strain JP-59 in human hepatoma cell line PLC/PRF/5. This manuscript is well written and seems important in this area. This reviewer has several minor comments.
1. In Materials and methods section, lines 93-94,
Authors wrote “...a total of 200 g of the fecal specimens was collected from a JP-59-infected rabbit on days 736.738 p.i….” Authors should describe how to infect in details.
2. It is difficult to understand why the authors choose PLC/PRF/5. In Discussion section, authors discuss about the cell culture system for HEV infection and describe the reasons why they chose this cell line.
3. In line 123, why did authors choose 1 hr as adsorbed periods. If it is 4 hr or 8 hr, can this virus replicate in PLC/PRF/5 cells?
4. In line 127, why did authors choose 36 oC but not 37 oC?
5. In Table 1, authors demonstrated several primer sets. Which does primer set have better sensitivity?
6. In Discussion section, “PLC/PRF/5 cells are not the best choice for this purpose, and a novel cell culture system for rabbit HEV is required.” Which cells did authors think are better cell lines?
Comments on the Quality of English LanguageZhang et al. reported that the low replication efficacy of a Japanese Rabbit HEV strain JP-59 in human hepatoma cell line PLC/PRF/5. This manuscript is well written and seems important in this area. This reviewer has several minor comments.
1. In Materials and methods section, lines 93-94,
Authors wrote “...a total of 200 g of the fecal specimens was collected from a JP-59-infected rabbit on days 736.738 p.i….” Authors should describe how to infect in details.
2. It is difficult to understand why the authors choose PLC/PRF/5. In Discussion section, authors discuss about the cell culture system for HEV infection and describe the reasons why they chose this cell line.
3. In line 123, why did authors choose 1 hr as adsorbed periods. If it is 4 hr or 8 hr, can this virus replicate in PLC/PRF/5 cells?
4. In line 127, why did authors choose 36 oC but not 37 oC?
5. In Table 1, authors demonstrated several primer sets. Which does primer set have better sensitivity?
6. In Discussion section, “PLC/PRF/5 cells are not the best choice for this purpose, and a novel cell culture system for rabbit HEV is required.” Which cells did authors think are better cell lines?
Author Response
Ms. viruses-2417822: "Low Replication Efficiency of a Japanese Rabbit Hepatitis E Virus Strain in the Human Hepatocarcinoma Cell Line PLC/PRF/5," by Zhang et al.
Responses to Reviewer 1:
Zhang et al. reported that the low replication efficacy of a Japanese Rabbit HEV strain JP-59 in human hepatoma cell line PLC/PRF/5. This manuscript is well written and seems important in this area. This reviewer has several minor comments.
- In Materials and methods section, lines 93-94,
Authors wrote "...a total of 200 g of the fecal specimens was collected from a JP-59-infected rabbit on days 736.738 p.i…."Authors should describe how to infect in details.
Response: Since the "JP-59-infected rabbit" was described in a previous study, we have added the reference with that description to the revised manuscript.
- It is difficult to understand why the authors choose PLC/PRF/5. In Discussion section, authors discuss about the cell culture system for HEV infection and describe the reasons why they chose this cell line.
Response: We have added the following text to the Discussion to address this comment:
"…because the PLC/PRF/5 is broadly susceptible to HEV infection and the cell culture of rabbit HEV has been established by using this cell line."
- In line 123, why did authors choose 1 hr as adsorbed periods. If it is 4 hr or 8 hr, can this virus replicate in PLC/PRF/5 cells?
Response: Our previous research confirmed that 1 hr is enough for the adsorption, and the extended periods did not exert a significant effect on the virus adsorption. In addition, when considering the toxic substances included in 10% stool suspension, 1 hr for the adsorbed period was appropriate.
- In line 127, why did authors choose 36°C but not 37°C?
Response: The replication of HEV in the cell culture is extremely slow and thus takes a long time. The cells were thus incubated at 36°C to prevent overgrowth.
- In Table 1, authors demonstrated several primer sets. Which does primer set have better sensitivity?
Response: Although the sensitivity is unknown, the amplification with all primer sets was satisfactory.
- In Discussion section, "PLC/PRF/5 cells are not the best choice for this purpose, and a novel cell culture system for rabbit HEV is required. "Which cells did authors think are better cell lines?
Response: Although we have no clear answer to this question, we are now investigating the cell line LLC-RK1 derived from rabbit for better replication. We added the following text to the Discussion:
"It is of interest to examine whether the cell lines derived from rabbit were susceptible to rabbit HEV infection."

Reviewer 2 Report
Comments and Suggestions for Authors
In the present manuscript, Zhang et al. have investigated the replication of a rabbit strain in a human hepatocarcinoma cell line. This paper is of interest for the field as HEV strains do not replicate efficiently in cell culture and it is difficult to develop cell culture systems to study the virus. Moreover, very few studies have assesses rabbit strains. Nevertheless, the present manuscript needs some improvements to clarify the experiments performed as detailed below.
- Line 42: it could be useful to specify that rat HEV belong to the Rocahepevirus genus.
- Line 63-64: The sentence (“To date … in Japan”) is not clear.
- Line 107: “by” is repeated.
- Line 164-166: Is it a commercial or an in-house test? Please specify.
- Line 174-175: More details on the NGS could be provided briefly (extraction of RNA ?, technology used for NGS, …).
- It is important to better clarify within the result section the numbers of cells that were infected for each infection experiment and the volume of viral suspension added for inoculation (or number of copies of HEV) or specify the MOI used (fig.2, 3 and line 232-234). The inoculum added could also be indicated in the figures (Day 0). Similar titers are obtained in Fig. 3 and Fig.4 at 4 d.p.i suggesting that cells were infected at the same moi in both experiments?
- Line 180-181: fecal suspension used for rbIM223L?
- Line 184-185: This sentence is misleading. The titer was determined in the supernatant and not intracellularly.
- Line 187 and 197: What do you mean by “sign”?
- Line 191 and 193: Bottle? Please specify.
- Line 195-199: Did you infect A549 in parallel and detect viral replication with rbIM2141L/rbIM223L in A549? It could be useful to add these data in Fig. 2.
- Line 197-199: Did you observe cytotoxicity with the stool suspensions?
- Line 200: “JP-59 replicates in PLC/PRF/5 cells when a purified high-titer virus inoculum/viral suspension was used”.
- Line 201: “purified high-titer JP-59 viral suspension”
- Line 211: this experiment was performed to determine whether cell passaging was affecting viral replication rather than cell passage ?
- How was the cells trypsinised? Were they diluted when passaged ?
- Figure 4: ELISA. Were control rabbits also analysed ? What is the OD cut off value for the ELISA ?
- Table 2: In which domains of ORF1 are the amino acid changes found ? Are the amino acid changes found present in rbIM223LR?
- Discussion: The discussion is quite short. Have other cell types than PLC/PRF/5 and A549 been tested (or in the literature)? Are there rabbit cell lines (hepatic?) that have been/could be tested? Does all rabbit strains cause persistent infection in experimentally infected-rabbit?
Comments on the Quality of English Language
Quality of English is fine. Minor editing is required (spelling mistakes, repeated words). A few sentences could be rephrased for clarity purpose.
Author Response
Ms. viruses-2417822: "Low Replication Efficiency of a Japanese Rabbit Hepatitis E Virus Strain in the Human Hepatocarcinoma Cell Line PLC/PRF/5," by Zhang et al.
Responses to Reviewer 2:
In the present manuscript, Zhang et al. have investigated the replication of a rabbit strain in a human hepatocarcinoma cell line. This paper is of interest for the field as HEV strains do not replicate efficiently in cell culture and it is difficult to develop cell culture systems to study the virus. Moreover, very few studies have assesses rabbit strains. Nevertheless, the present manuscript needs some improvements to clarify the experiments performed as detailed below.
Line 42: it could be useful to specify that rat HEV belong to the Rocahepevirus genus.
Response: We have added the following text:
"…and rat HEV belong to the Rocahepevirus genus."
Line 63-64: The sentence ("To date … in Japan") is not clear.
Response: We revised the text as follows:
"A rabbit HEV strain, JP-59, was identified in a feral rabbit in Japan."
Line 107: "by" is repeated.
Response: We have corrected that mistake.
Line 164-166: Is it a commercial or an in-house test? Please specify.
Response: We have specified that the ELISA is an "in-house" test in the revised manuscript.
Line 174-175: More details on the NGS could be provided briefly (extraction of RNA?, technology used for NGS, …).
Response: More details of the NGS method were added.
It is important to better clarify within the result section the numbers of cells that were infected for each infection experiment and the volume of viral suspension added for inoculation (or number of copies of HEV) or specify the MOI used (fig.2, 3 and line 232-234). The inoculum added could also be indicated in the figures (Day 0).
Response: The numbers of cells (5×105 cells/bottle) used for each infection experiment were clarified in section 2.3. Cell culture and virus infection.
The volume of the viral suspension used for inoculation (1 mL) was added in the revised manuscript:
"To isolate the JP-59 by using PLC/PRF/5 cells, we inoculated the cells with 1 mL of JP-59s, a stool suspension derived from a persistently JP-59-infected rabbit that contains 1.1×107 copies/mL viral RNA. Triplicate samples were used for the inoculation. 1 mL of LRs, the virus suspension derived from an rbIM223LR-infected rabbit, contains same RNA copy number, was used as a positive control. "
Similar titers are obtained in Fig. 3 and Fig.4 at 4 d.p.i suggesting that cells were infected at the same moi in both experiments?
Response: We do not agree with this suggestion. Please note too that Figure 3 showed the results of the replication of strain JP-59 in PLC/PRF/5 cells and Figure 4 showed the results of the infectivity of IP-59c in the rabbits.
Line 180-181: fecal suspension used for rbIM223L?
Response: The fecal suspension, LRs, derived from an rbIM223LR-infected rabbit, was used to inoculate PLC/PRF/5 cells and was used as the positive control.
Line 184-185: This sentence is misleading. The titer was determined in the supernatant and not intracellularly.
Response: We revised the sentence as follows:
"In contrast, the viral RNA in the supernatants of the JP-59s-infected cells decreased rapidly…"
Line 187 and 197: What do you mean by "sign"?
Response: We revised the relevant text as follows:
"no evidence of the virus replication was observed."
and
"but we observed none of the evidence of virus replication in the case of JP-59."
Line 191 and 193: Bottle? Please specify.
Response: We added the following sentence to the Figure 2 legend:
"PLC/PRF/5 cells were cultured in 25-cm2 bottles.
Line 195-199: Did you infect A549 in parallel and detect viral replication with rbIM2141L/rbIM223L in A549? It could be useful to add these data in Fig. 2.
Response: We have two strains, rabIM223LR and JP-59, and we were unable to investigate the replication of rbIM2141L/rbIM223L strains in A549 cells.
Line 197-199: Did you observe cytotoxicity with the stool suspensions?
Response: No clear cytotoxicity was observed when inoculation with the stool suspensions derived from JP-59 and LRs was used.
Line 200: "JP-59 replicates in PLC/PRF/5 cells when a purified high-titer virus inoculum/viral suspension was used".
Response: That text was revised as follows:
"JP-59 replicates in PLC/PRF/5 cells when a purified high-titer virus was used"
Line 201: "purified high-titer JP-59 viral suspension"
Response: To distinguish it from the stool suspension, we would like to use "purified high-titer JP-59".
Line 211: this experiment was performed to determine whether cell passaging was affecting viral replication rather than cell passage?
Response: In general, the infected cells were maintained by changing the medium without the cell passage. Since the viral RNA detected in the cell culture supernatants was quite low, we passaged the JP-59F6-infected cells to determine whether the cell passage affects the virus replication.
How was the cells trypsinised? Were they diluted when passaged?
Response: We added the following description:
"The cells were washed three times with PBS and incubated with 1 mL of 0.05% trypsin at 37°C for 10 min. After the trypsin was removed, the cells were diluted with 20 mL of the maintenance medium, separated into two bottles, and incubated at 36°C. "
Figure 4: ELISA. Were control rabbits also analysed? What is the OD cut off value for the ELISA ?
Response: Since the rabbits used in this study were negative for rabbit HEV, the replication of the virus before and after the virus inoculation was compared. No control rabbits were used. The cutoff value for the anti-HEV IgG was 0.151, and a description of this was added to section 2.6. in the revised manuscript.
Table 2: In which domains of ORF1 are the amino acid changes found? Are the amino acid changes found present in rbIM223LR?
Response: We added the following description:
"The mutation Thr566Ile was observed in the papain-like cysteine protease (PCP) domains and a mutation Gly749Ser was located in the hypervariable region (HVR) domain of ORF1. The amino acid, 566Ile, was identified in rbIM223LR, but 749Ser and 4Phe were not present in rbIM223LR."
Discussion: The discussion is quite short. Have other cell types than PLC/PRF/5 and A549 been tested (or in the literature)? Are there rabbit cell lines (hepatic?) that have been/could be tested? Does all rabbit strains cause persistent infection in experimentally infected-rabbit?
Response: As far as we know, no other cells including rabbit cell lines have been tested for rabbit HEV replication. Since we were unable to obtain the rabbit hepatic cell line, we are using LLC-RK1 cells (a rabbit kidney cell line) to see whether the cells are susceptible to rabbit HEV infection.
We added the following description:
"It is of interest to examine whether the cell lines derived from rabbits were susceptible to rabbit HEV infection."
Although we do not know whether all rabbit HEV strains cause persistent infection, two strains (rbIM223LR and JP-59) caused persistent infection in experimentally infected rabbits.
